# Assessment of Pollutants from Diffuse Pollution through the Correlation between Rainfall and Runoff Characteristics Using *EMC* and First Flush Analysis

**Maria Elisa Leite Costa** *, **Daniela Junqueira Carvalho** and **Sergio Koide**

Civil and Environmental Engineering Department, University of Brasilia, Federal District,
Brasília 70910-900, Brazil; d.junqueirac@gmail.com (D.J.C.); skoide@unb.br (S.K.)
* Correspondence: mariaelisaleitecosta@hotmail.com

**Abstract:** Urban stormwater runoff is an important source of pollution in receiving water bodies, mainly in cities in development. However, strategies to deal with the impacts caused by the runoff discharges, such as implementing a sustainable urban drainage system (SUDS) with optimized management, need information usually obtained through monitoring studies. Brasília is a city that has one of the highest urban growth rates in Brazil, with significant impacts on urban water resources, including diffuse pollution, generated by new unregulated urban developments that initially start being built with precarious sanitation infrastructure. The Vicente Pires (VP) watershed is highly urbanized and comprises two areas that have been intensively occupied more recently, at a fast pace, and do not have yet basic sanitation systems fully implemented. Stormwater quality at the outlet of the VP watershed was analyzed by monitoring the rainfall, runoff flows, and pollutant concentration. Event Mean Concentration (*EMC*) and first-flush (FF) phenomenon were calculated, and hydrologic characteristics were compared for different events through correlation analysis. During dry periods the flow varied between 0.5 and 1.29 $m^3/s$, while in flood periods the maximum value was 72.17 $m^3/s$, forming floods with great volume. Nitrate during dry periods stands out with its high concentration; the maximum was 1.49 mg/L, while the maximum concentration during the flood events was 0.43 mg/L, probably due to dilution. Ammonia results showed very low values, probably because nitrification is occurring up to the collection point. The *EMC* values of solids in flood events were higher and can be attributed to river bed scour along the VP watershed. The *EMC* SS values for the VP watershed are also similar to areas in the initial stages of building development. The *EMC* values in the dry season indicate strong correlations between some water quality parameters such as $NH_3^+$-N and SS, TS and $NO_3^-$-N; $NO_3^-$-N, and COD. These correlations indicate that these pollutants are probably being generated by the same source, probably sewage discharges. During flood events, the correlation between pollutant loads and peak flow can be associated with the scouring during surface washing off, because greater concentrations of solids and organic matter occur in events with greater flow rates. For the first 30% of the initial runoff volume, about 29% of SS, 38% of $NH_3^+$-N, and 35% of reactive P were carried during flood events. It was verified that large values of maximum or mean rainfall intensity are related to the occurrence of First Flush (FF) for most pollutants. Antecedent dry days (ADD) did not influence build-up processes in this watershed; however, they are related to FF occurrence. Data indicate that the sewage and stormwater collection networks were being installed caused a high impact on observed water quality, with high concentrations of solids during flood events. On the other hand, the wastewater collection after the sewer network installation led to a decrease in COD concentrations over time. For sustainable management of diffuse pollution, the adoption of distributed SUDS to enhance runoff volume reduction is a recommended solution for the case.

**Keywords:** urbanization; water quality; stormwater

## 1. Introduction

Urbanization is a global phenomenon that can impart significant changes in hydrologic [1–3] and water quality systems [3–7]. Urban growth affects directly drainage processes due to vegetation removal and replacement of pervious areas by impervious surfaces which increment flows and pollutant loads that are washed downslope during storms, causing the increase of diffuse pollution loads and floods, inducing water quality deterioration [8–16]. Urban stormwaters are recognized as a source of diffuse pollution that carry a wide range of pollutants, including particulates and dissolved substances, accumulated due to land use [17,18]. In Brazil, the sewerage and storm drainage systems are separated and it is expected that stormwater presents very distinct quality and quantity characteristics from domestic sewage [1].

Strategies to deal with stormwater are needed to construct drainage systems, in order to ensure their functionality to guarantee flood control, public health, and economic sustainability [1,19] nowadays, using sustainable urban drainage systems [20].

Information on stormwater characteristics may be achieved through monitoring studies [5,7,9,12]. Usually, pollutant loads are often underestimated [21] due to difficulties in monitoring, mainly during flood events [22]. The pollutant loads are aggravated by deficient individual septic systems usually adopted prior to implementation of sewage collection systems and by deficient solid waste collection services.

Runoff characteristics of each event monitored provide information to support the implementation of compensatory techniques for urban drainage that may reduce this type of pollution [23]. In addition, stormwater runoff pollution, primarily in the first minutes after the start of a rainfall event, was comparable to or greater than sewage pollution (evaluated by monitoring the VP River water quality during the dry periods) [24], which is in accordance to the first flush concept in which the first part of a rainfall event carries the major part of the pollutant load [23], which indicating how SUDs should be designed and implemented, so FF can be used to locate and size SUDs for optimal efficiency.

Brasília, Brazil's capital, is a city that is experiencing large urban growth, with a population increase of more than 4.8% in the 2015–2018 period [25]. Therefore, Administrative Regions (ARs) such as Vicente Pires and Arniqueiras, which were considered rural areas in the mid-1990s, became irregular urban settlements that are presently being established as legal urban areas.

The Vicente Pires (VP) River runs between both ARs cited and has become an important urban watershed which discharges into the Riacho Fundo River, a major tributary of the Paranoá Lake. In the 1970s and 1980s, the lake was eutrophicated and extensive works were carried out to improve the sewage collection and to treat wastewater to a tertiary level. By 1998, the lake recovered to a good condition, but over the last decade the water quality has been worsening again. This lake's water has been used for domestic water supply since 2018, when Brasilia suffered a huge water supply crisis, including the implementation of water supply restrictions [26]. The environmental sustainability of the lake is significantly affected by the VP River nutrient loads and sediments produced by the urban area and discharged in this water body, which has suffered riverbed silting [27–29]. In spite of that, official water quality monitoring programs set up only conduct monthly water sampling, with no event-based monitoring.

The aims of this research were as follows:

(1) to compare the Vicente Pires River water quality in flood events using the event mean concentration (*EMC*) and also by the first flush phenomenon, estimated based on multiple monitoring datasets.
(2) to correlate the pollutant load with runoff and rainfall characteristics.
(3) to discuss the possible sources of pollutants.

## 2. Materials and Methods

### 2.1. Study Area—Watershed Characteristics

The study area is an urban watershed in the city of Brasilia, Brazil. The city is located in the Federal District (DF), which is the smallest of Brazil's 27 federation units, at approximately 5800 km$^2$. Brasília is located in the region called the Brazilian Central Plateau, in the Cerrado biome, where the natural vegetation includes different types of vegetation, mainly of savannah type [30]. In the Federal District, the predominant soils are Red-Dark and Yellow Red Latosols, well-drained with moderate infiltration capacity and high porosity, favoring groundwater recharge, and Cambisols, being less permeable [31,32]. However, urban soils usually are altered by urbanization activities. According to the Köppen classification, the regional climate is the Savannah Climate (Aw), with a dry season during May–September and a rainy season during summer (October–April), with annual average precipitation of 1500 mm [33].

Because it is situated on a high plateau, the Federal District is considered a headwater region. Three major water basins in Brazil have origins within the DF. One of the Federal District's most important water bodies is the Paranoá Lake.

The study area is the Vicente Pires Watershed, which is inserted in the Paranoá Lake Watershed and has 90 km$^2$ (Figure 1) and a population of about 350,000 people [25]. The topography is predominantly softly undulated, with the terrain being steeper near the springs of the watercourses, which are located in a plateau border region [34]. Elevations vary from 1007 m to 1250 m and the average slope is 7.3%. Land uses in the watershed area are mostly urban. According to the city's Land Use Master Plan, just a small portion of the area (approximately 14%) is designated as a rural zone, and the urban zones, consolidated and in expansion, occupy the other part [35]. In terms of land cover, 52.2% of the area is occupied by buildings and roads, 37.5% by vegetation, 8% by exposed soil, and 2% by agriculture. The Vicente Pires River is the main water body and discharges into the Riacho Fundo River, comprehending the more urbanized and densely populated affluent of the Paranoá Lake [36–38].

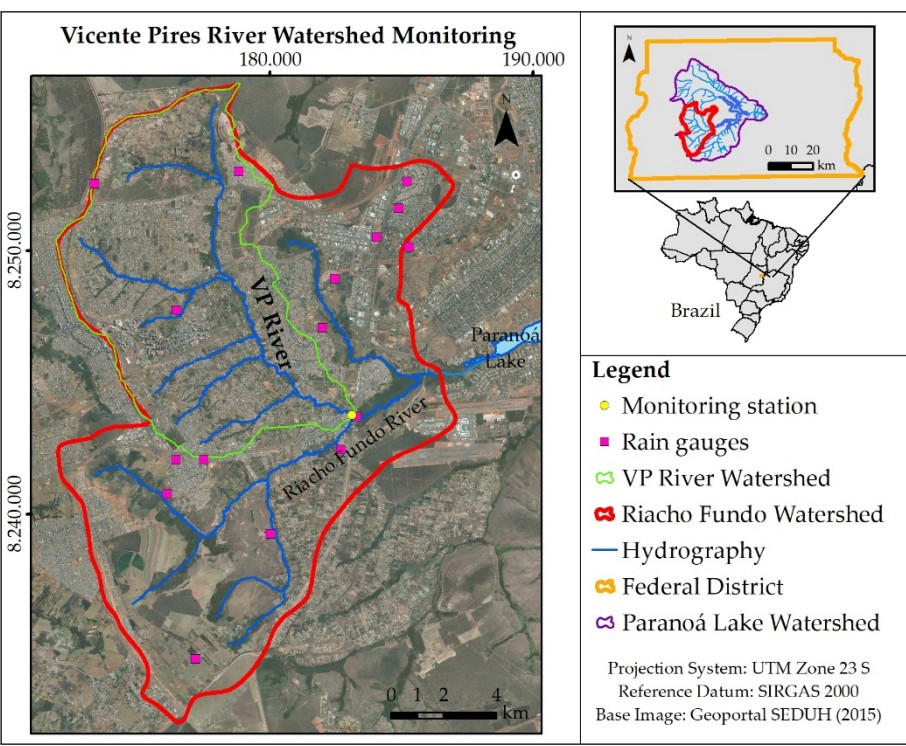

**Figure 1.** Location of the study area: Vicente Pires Watershed in the Federal District, Brazil.

Brasília is known as a planned city; however, the plan comprises only part of the territory and several originally unplanned occupations occurred around it, causing many problems. Additionally, many allotments were developed in farmland areas, and in the VP watershed the rural area settlements, called Vicente Pires and Arniqueiras, were occupied by irregular urbanized areas without sanitation infrastructures, such as water supply, sewage collection, and urban drainage networks. Consequently, flood events are quite common. After many years of vindication, land regularization and construction of such infrastructure are underway.

The Federal District has legal instruments for stormwater and water resource management, including municipal plans specific to these topics. The sewer rate is based on water usage and is charged in the water bill, but there are no stormwater management taxes yet. Local agencies are conducting studies for the establishment of these taxes in the DF.

In the process of legalization of the urban occupation in the VP area, the urban drainage system under construction will include, in addition to the drainage network, the construction of 22 detention ponds to improve the sustainability of the receiving water bodies. The Arniqueiras area also does not have the sanitary infrastructure, but it is still under land regularization; therefore, there is no prediction about the beginning of stormwater management infrastructure implementation.

### 2.2. Field Data Acquisition System and Water Sample Analyses

Eleven events were monitored in terms of rainfall, flow, and water quality; six of them occurred during the rainy season and five during the dry season. The river stage was monitored using a pressure transducer level logger (model WL6, GlobalWater—Yellow Springs, OH, USA) and water sampling was carried out using an autosampler (ISCO 6712, Teledyne—Thousand Oaks, CA, USA). Both pieces of equipment were installed at the outlet of the VP watershed from 2018 to 2020. Flow rates were measured using a river discharge measurement system (River Surveyor M9, Sontek—San Diego, CA, USA). In the rainy season, samples were collected during the flood events with a 10 min time step, while in the dry season the sampling was carried out with 1 h intervals for a 24 h period. Rainfall intensity was monitored by four rain gauges in the study area with a 5 min time step.

Using a rating curve, flow rates and runoff volumes were estimated for evaluation of the diffuse pollution loads generated by the build-up and wash-off processes on the VP watershed. The runoff volume was accounted for a period of up to 4 h during each rainfall event while the water sampling was occurring. During the dry season, it was accounted for over a period of 24 h. The runoff coefficient was calculated by dividing the runoff volume at the outlet (estimated by hydrograph separation) by the rainfall volume over the area (Thiessen method).

Water samples were analyzed in the Laboratory of Environmental Sanitation of the University of Brasilia for chemical oxygen demand (COD) as an indirect measure of organic matter content, nutrients (nitrogen and phosphorus in the forms of nitrite ($NO_2$-N), nitrate ($NO_3$-N), ammonia ($NH_3$-N), total phosphorus (TP) and reactive phosphorus (RP)), and sediments (total solids (TS), suspended solids (SS) and dissolved solids (DS)). Analyses were carried out according to the Standard Methods for the Examination of Water and Wastewater [39]. Due to the large number of samples collected per event and the sampling frequency, automated methods were selected when available. Table 1 presents the applied methods and their technical specifications, as well as the reference methods from Standard Methods (SM) that correspond to them

**Table 1.** Methods, equipment, and ranges for the analysis of water quality.

| Parameter | Method | SM Reference Method | Equipment Used | Detection Range |
|---|---|---|---|---|
| COD | Reactor digestion | 5220 D | Spectrophotometer HACH DR 2010 and Digestion reactor | 0–150 mg COD/L |
| NO$_{-2}$-N | Diazotization | 4500-NO2 B | Spectrophotometer HACH DR 4000 | 0–0.3 mg NO$_{-2}$-N/L |
| NO$_{-3}$-N | Cadmium Reduction | - | Spectrophotometer HACH DR 4000 | 0–5 mg NO$_{-3}$-N/L |
| NH$_3$-N | Nesslerization | 4500-NH3 C (1995) | Spectrophotometer HACH DR 4000 | 0–2.5 mg NH$_3$-N/L |
| TP | Ascorbic acid with acid persulfate digestion | 4500-P F-H | Spectrophotometer HACH DR 4000 and Digestion reactor | 0–3.5 mg PO$_4^{3-}$/L 0–1.1 mg P/L |
| RP | Ascorbic acid | 4500-P F-G | Spectrophotometer HACH DR 4000 | 0–2.5 mg PO$_4^{3-}$/L |
| TS | Gravimetric | 2540 B | Analytical balance Adventurer—OHAUS | 0–210 g/L |
| SS | Gravimetric | 2540 D | Analytical balance Adventurer—OHAUS | 0–210 g/L |
| DS | Differential | - | - | - |

*2.3. Water Quality Assessment Modelling*

The *EMC*—event mean concentration—for each parameter was calculated according to Equation (1) to compare the monitoring results. *EMC* is measured as the ratio of total pollutant mass and total runoff flow used to quantify a single event through composed sampling. It is a trustable parameter and allows easier comparison between events and different locations [40–43].

$$EMC = \frac{\sum_i^n (Qi \cdot Ci) \cdot \Delta t}{\sum_i^n Qi \cdot \Delta t} \tag{1}$$

With:
*EMC* = Event Mean Concentration (mg/L).
$Q$ = flow rate (m$^3$/s).
$C$ = pollutant concentration (mg/L).
$\Delta t$ = time intervals.

The *EMC* is appropriate for evaluating the effects of runoff on receiving water bodies because it is considered an index of the polluting potential of the event [16]. The *EMC* calculation was applied for all parameters and monitored events to make the investigation clearer and to compare the data between events.

In diffuse pollution, due to pollutant accumulation and washing directly related to rainfall, there can be an increase in pollutant load in the first portion of the flow, known as "first flush" [44–46]. To identify this phenomenon, the dimensionless curve M(V) that indicates the disproportionality of concentrations or mass during the first portions of surface flow volume [47,48] is used, which were determined for the monitored events in VP River, calculated with Equation (2).

$$\frac{\sum_{i=1}^{j} C_i Q_i \Delta t_i}{\sum_{i=1}^{N} C_i Q_i \Delta t_i} = f\left(\frac{\sum_{i=1}^{j} Q_i \Delta t_i}{\sum_{i=1}^{N} Q_i \Delta t_i}\right) = f\left(\frac{\sum_{i=1}^{j} V_i}{\sum_{i=1}^{N} V_i}\right) \tag{2}$$

where $C$ is the concentration of pollutants in the sample, $Q$ is the flow rate through the cross-section of the river, $N$ is the total number of samples, and $V$ is the volume discharged during the time interval $\Delta t_i$. This way, on the *X*-axis is the accumulated fraction of water volume and on the *Y*-axis is the accumulated fraction of pollutant load carried by the accumulated volume.

The numerical coefficient "b" of the curves (Equations (3) and (4)), was also calculated using simple linear regression [49]. This value expresses the length between the curve M(V) and the 45° line (bisector). If it is equal to one, pollutant carriage is uniform. Values lower than one indicate the occurrence of the first flush. The lowest the value of b is, the greater is the value and, therefore, the pollutant load carried in the first volumes.

$$F(X) = X^b \tag{3}$$

$$F(X) = X^b \langle = \rangle ln(F(X)) = b \cdot ln(X) \tag{4}$$

where $X$ is the accumulated volume and $F(X)$ is the accumulated load; $X \in [0, 1]$, $F(0) = 0$ and $F(1) = 1$. It is usually considered that the experimental adjustment between M(V) and $F(X)$ is satisfactory, with correlations of $R^2 > 0.9$.

### 2.4. Data Analysis

Multivariate analyses, such as correlation, were performed to analyze the data collected, using two pieces of software: PAST 3.26 and ADDIn XLSTAT in Excel—Microsoft 365. All the analyses were performed for two separated periods: dry and flood. The correlation evaluation numerically shows the linear relationship between variables through the Pearson's coefficient (r), for which a value of r < 0 indicates an inversely proportional correlation, and a value of r > 0 indicates a direct correlation, where the closer the value is to 1 the stronger the relationship, and the closer the r value is to 0, the weaker the correlation is [50].

System optimization requires that research studies be turned into concrete applications; as such, it is important to look at the benefits to the management of the system from the findings of these studies, as done by [51,52]. Based on the results of the water quality assessment of the VP River in the dry and rainy periods, this study also conducted a reflection on the perspectives for using the data collected in the management of stormwater and water resources in the Federal District, pointing out some managerial insights.

## 3. Results and Discussion

### 3.1. Rainfall and Runoff Analysis

The event's characteristics are shown in Table 2. It is noted that the rainfall characteristics are remarkably diverse and the maximum rainfall intensity occurred on 3 April 2018, 39.40 mm/h, with a higher average intensity of 14.39 mm/h. However, this event did not present a high peak flow either a high runoff coefficient, which may have happened due to the variability of rainfall distribution along the watershed, or high infiltration rates as a result of the soil being dry, given that the event had five ADD (antecedent dry days).

**Table 2.** Rainfall and Runoff characteristics for event monitored.

| Event | Average Vol. Rainfall | Average Intensity Rainfall | Max Intensity Rainfall | ADD | Peak Flow | Runoff Vol. | Runoff Coef. |
|---|---|---|---|---|---|---|---|
| 1 August 2018 | - | - | - | 74 | 1.29 | 85,003.96 | - |
| 1 June 2019 | - | - | - | 13 | 0.9 | 73,936.94 | - |
| 19 June 2019 | - | - | - | 31 | 0.81 | 68,362.27 | - |
| 12 August 2019 | - | - | - | 81 | 0.8 | 59,201.82 | - |
| 29 August 2019 | - | - | - | 101 | 0.5 | 41,841.93 | - |
| 19 February 2018 | 14.33 | 9.01 | 24.00 | 1 | 40.94 | 258,735.72 | 0.20 |
| 4 March 2018 | 17.20 | 14.39 | 39.40 | 5 | 9.14 | 347,488.35 | 0.32 |
| 13 March 2018 | 4.40 | 7.64 | 10.40 | 4 | 10.7 | 151,830.15 | 0.38 |
| 15 November 2018 | 17.40 | 5.04 | 8.88 | 5 | 6.81 | 64,541.48 | 0.04 |
| 18 February 2019 | 12.10 | 5.40 | 11.70 | 2 | 72.17 | 321,776.75 | 0.30 |
| 2 January 2020 | 32.33 | 9.16 | 15.85 | 1 | 53.96 | 346,177.43 | 0.12 |

In the dry period, peak flows are low due to the lack of rainfall, which causes runoff from all surfaces in the watershed. The runoff volume displayed in Table 2 for the dry season events is related to the base flow during one day (24 h monitoring). It was observed that the greater the ADD number is, the less water is available from the aquifers, except for the event on 1 June 2019. During the flood events, the maximum peak flow was 72.17 m$^3$/s, and the maximum runoff volume reached 347,488.35 m$^3$ for a single monitored event, which represents five times the average for the base flow volume measured over 24 h in the dry season, but occurring in only a few hours (4 h flood wave). Therefore, it is possible to compare the peak flow monitoring results in Figure 2, where the axes show how high

the flows are during a flood event, which can have many impacts on the rivers, such as erosion and diffuse pollution.

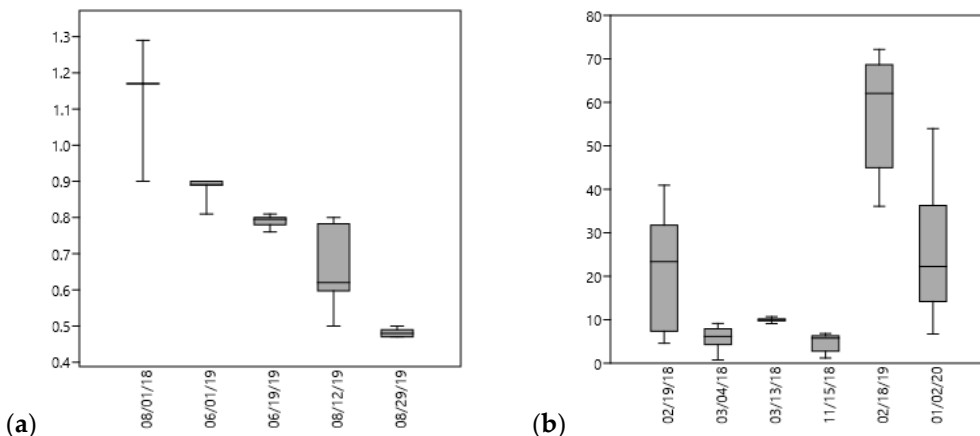

**(a)**          **(b)**

**Figure 2.** Flow distribution (m$^3$/s) in each monitoring event, separated into dry (**a**) and flood (**b**) events.

### 3.2. EMC Analysis

In Table 3 it was observed that the *EMC* for phosphorus, RP, and TP reached their maximum values of 0.07 and 0.15 mg/L, respectively, lower than the ones found in the urban direct runoff [10]. Even so, the TP and RP *EMC* values are similar to those found in urban areas with controlled domestic sewage, which range from 0.01–1.71 for TP and 0.026–0.48 for RP [53–57]. This, however, does not reflect the reality of the whole study area, where there is no full coverage of sewage collection or treatment, but rather the widespread use of septic tanks.

**Table 3.** *EMC* estimates for each event at VP River in mg/L.

| | Event | RP | TP | NO$_2$-N | NO$_3$-N | NH$_3$-N | SS | TS | DS | COD |
|---|---|---|---|---|---|---|---|---|---|---|
| DRY SEASON | 1 August 2018 | 0.05 | 0.09 | 0.01 | 1.16 | 0.05 | 1.49 | 76.01 | 74.54 | 12.40 |
| | 1 June 2019 | 0.03 | 0.04 | 0.01 | 1.49 | 0.24 | 14.77 | 78.31 | 60.67 | 32.05 |
| | 19 June 2019 | 0.02 | 0.06 | 0.02 | 1.17 | 0.25 | 11.93 | 98.02 | 86.09 | 22.30 |
| | 12 August 2019 | 0.07 | 0.02 | 0.03 | 0.79 | 0.08 | 3.83 | 51.21 | 50.60 | 8.70 |
| | 29 August 2019 | 0.05 | 0.11 | 0.01 | 1.10 | 0.17 | 5.31 | 108.17 | 97.29 | 6.88 |
| FLOODS | 19 February 2018 | 0.03 | 0.10 | 0.02 | 0.16 | 0.18 | 2919.23 | 3492.01 | 1084.41 | 297.38 |
| | 4 March 2018 | 0.02 | 0.05 | 0.01 | 0.12 | 0.12 | 1721.44 | 2984.78 | 1185.82 | 238.32 |
| | 13 March 2018 | 0.02 | 0.02 | 0.01 | 0.21 | 0.08 | 221.13 | 389.05 | 76.20 | 34.78 |
| | 15 November 2018 | 0.07 | 0.15 | 0.00 | 0.17 | 0.13 | 1132.53 | 1411.95 | 379.89 | 116.12 |
| | 18 February 2019 | 0.04 | 0.10 | 0.01 | 0.08 | 0.14 | 1767.49 | 4130.12 | 2362.63 | 382.05 |
| | 2 January 2020 | 0.02 | 0.09 | 0.01 | 0.43 | 0.33 | 2666.36 | 2991.09 | 350.44 | 341.74 |

Regarding nitrogen, it was observed that during dry periods the nitrate stands out with high concentrations, as the maximum NO$_3$-N *EMC* found was 1.49 mg/L for a dry season event. Comparing to the maximum concentration during a flood event, 0.43 mg/L, it is concluded that this is probably due to the dilution caused by the large volume of surface runoff transported during floods. The nitrate *EMC* values are similar to the results obtained for predominantly residential areas, in suburban regions larger than 40 ha [58]. Meanwhile, ammonia results showed very low values when compared with the data from the Jardim Vista Alegre watershed in São Paulo, also an urbanized watershed [53], probably due to the nitrification that occurred up until the collection point in the VP River, and it also presents similarities with construction areas in the USA in the stage of clearing and grubbing [58], exactly the stage of installation of stormwater drainage systems at the moment of data collection in the VP watershed.

The *EMC* values for solids, in general, are higher at the higher flow rates and are strongly influenced by runoff. Even during flood events, with flows as low as 10 m$^3$/s, there was a great variation of *EMC* for solids. However, it is still higher than any event that happened during the dry period. The *EMC* values for flood events were higher than those found for runoff of surfaces such as roofs (29 mg/L), terraces (490 mg/L), and streets (498 mg/L) [59]. These *EMC*s can be attributed to erosion phenomenon on the banks of the river in the VP watershed, a fact also identified in the Saquarema stream in the city of Belo Horizonte [60]. The *EMC* values for SS obtained for the VP watershed are also similar to those of areas in the USA where the predominant land use and cover correspond to areas under construction during the initial stages, such as clearing and grubbing [58].

During the flood events, the *EMC* values for COD were higher than in the dry periods, with a maximum of 382.05 mg/L and 32.05 mg/L, respectively. The *EMC* values are similar to those of Saquarema stream in flood periods, 87–340 mg/L, even when at the dry period the *EMC* for COD increased [60], which was not verified in the VP River, due to the relative control of sewage discharge during the dry period, when the septic tanks work properly.

Furthermore, *EMC* values for COD are lower than the ones found in the Jardim Vista Alegre (281.34–874.18 mg/L) and Campos Lemos (231.12–644.72 mg/L) watersheds [53], probably because of a minor discharge of untreated sewage into the VP River. Gromaire-Mertz et al., (1999) [59] investigated the *EMC* for different surfaces and reported that the site that produces the higher *EMC* for COD is streets, with an average of around 131 mg/L, followed by terraces (95 mg/L), and lastly roofs (31 mg/L), corroborating that the washing off of these surfaces provides a large discharge of organic matter, not only wastewater from households.

It was observed that runoff volume dilutes the nutrients (nitrogen and phosphorus) because of the large increase in the available volume. In the dry period, in addition to constant flow rates or with exceptionally low variations, all parameters also show steady concentrations, representing the water quality characteristics of the base flow. It is important to point out that when a non-regular variation had been observed, then it would not have been caused by a point source discharge, such as a large effluent discharge from a wastewater treatment plant, for example. Thus, it can be assumed that if a discharge is occurring, it is an illegal, diffuse source, and continuous over time.

Huang et al., (2007) [61] investigated the characteristics of runoff in an urban basin in Macau and found that the greatest rainfall with the highest number of antecedent dry days (ADD) had the highest *EMC* for nitrogen, suspended solids, and organic matter. Therefore, the influence of hydrological parameters on monitored water quality was also analyzed for each season studied.

The *EMC* for the dry season demonstrated strong correlations between the quality parameters (Figure 3). The correlation coefficients found were as follows: r = 0.93 for $NH^{+3}$-N and SS; r = 0.72 for $NH^{+3}$-N and COD; r = 0.57 for $NH^{+3}$-N and TS; r = 0.64 for $NH^{+3}$-N and $NO^{-3}$-N; and r = 0.82 for $NO^{-3}$-N and COD. These values indicate that these pollutants probably came from the same source, which, during the dry season, might be associated with sewage discharges. TP presented a correlation with DS (r = 0.84) and TS (r = 0.71), indicating that it is associated with sediments. Comparing *EMC* values with hydrological data, the connection of baseflow volume in the cross-section with COD (r = 0.50) and with $NO^{-3}$-N (r = 0.40) is noted, indicating that the greater the volume crossing the monitored section is, the greater is the concentration of pollutants.

Analyzing the *EMC* correlation with hydrological data of the flood events (Figure 4), it is noted the importance of rainfall in the water quality of the VP River. The greater the average rainfall volume, the higher the *EMC* values for $NH^{+3}$-N (r = 0.90), $NO_{-3}$-N (r = 0.72), SS (r = 0.65) and COD (r = 0.52). COD and SS are strongly correlated, as well as ammonia and nitrate, which can be attributed to possible discharges from overflowing septic tanks that have a limited capacity and, with abnormal rainfall events, can discharge their exceeding volume directly into the stormwater drainage system.

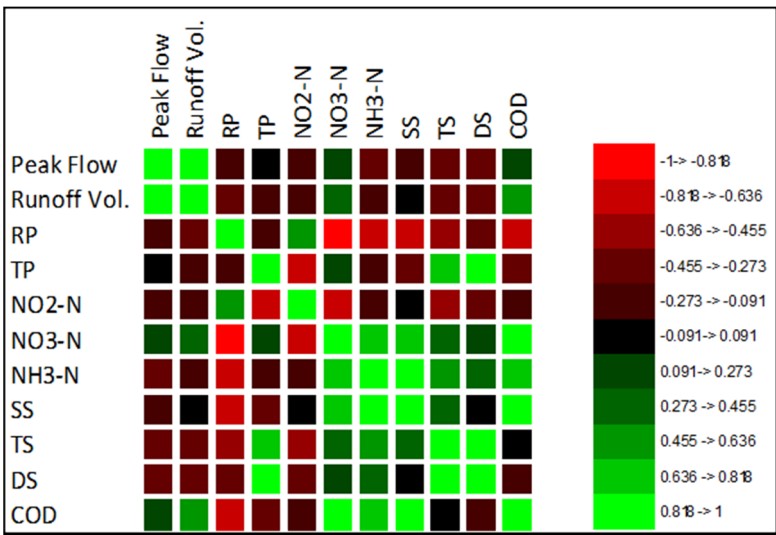

**Figure 3.** Correlation between the hydrological and the water quality characteristics (*EMC*) during the dry period.

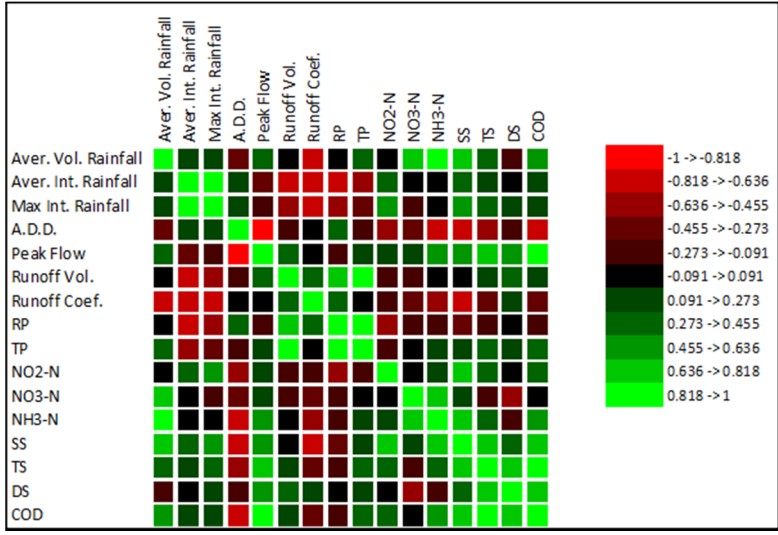

**Figure 4.** Correlation between the hydrological data and the water quality characteristics during floods.

Regarding diffuse pollution, the correlation between pollutants and peak flow is an analysis that can be associated with the drag force during surface washing. It was observed that the strongest peak flow correlations were with COD (r = 0.86), TS (r = 0.74), and DS (r = 0.62), corroborating that the highest concentrations of solids and organic matter occur in events with greater flow rates. The greatest volumes of surface runoff flow resulted in greater *EMC* values for RP (r = 0.765) and TP (r = 0.89), probably because this flow suspends sediments at the bottom of the riverbed, releasing the pollutants associated with them.

The number of antecedent dry days (ADD) is often analyzed in the scientific literature because it is directly associated with diffuse pollution due to the phenomenon known as build-up, the accumulation of pollutants on the surface. In the present study, it was observed that the greater the ADD number, the lower the peak flow (r= −0.84). This is associated with increased infiltration, due to the fact that dry soil is able to retain more rainfall volume, which is why ADD do not influence discharged pollution as much since the reduction in surface flow causes the loads discharged in the VP River to also decrease.

The present study did not find any high value of *EMC* for SS correlation with ADD, rainfall duration, volume, or intensity, a result similar to the experimental study developed

in a residential watershed in France [59]. However, when associating SS concentrations with hydrological parameters, the strongest correlation corresponds to that with the average volume of rainfall, emphasizing that the concentration is associated with the flow rate, i.e., diffuse pollution. Park et al., (2019) [62] also calculated correlations between hydrological and water quality parameters obtaining high correlation values between total rainfall volume and intensity, and between flow rate and SS, as seen in the VP River.

### 3.3. First Flush Analysis

The behavior of the first flush is quite heterogeneous when comparing the dry and flood season events, and even within the same event for the different pollutants. The event on 19 February 2018 presents all curves above the 45° bisector, except for nitrite. That result diverges from those on 1 August 2018, 18 February 2019 and 2 January 2020, where most curves are situated below the bisector (Figure 5).

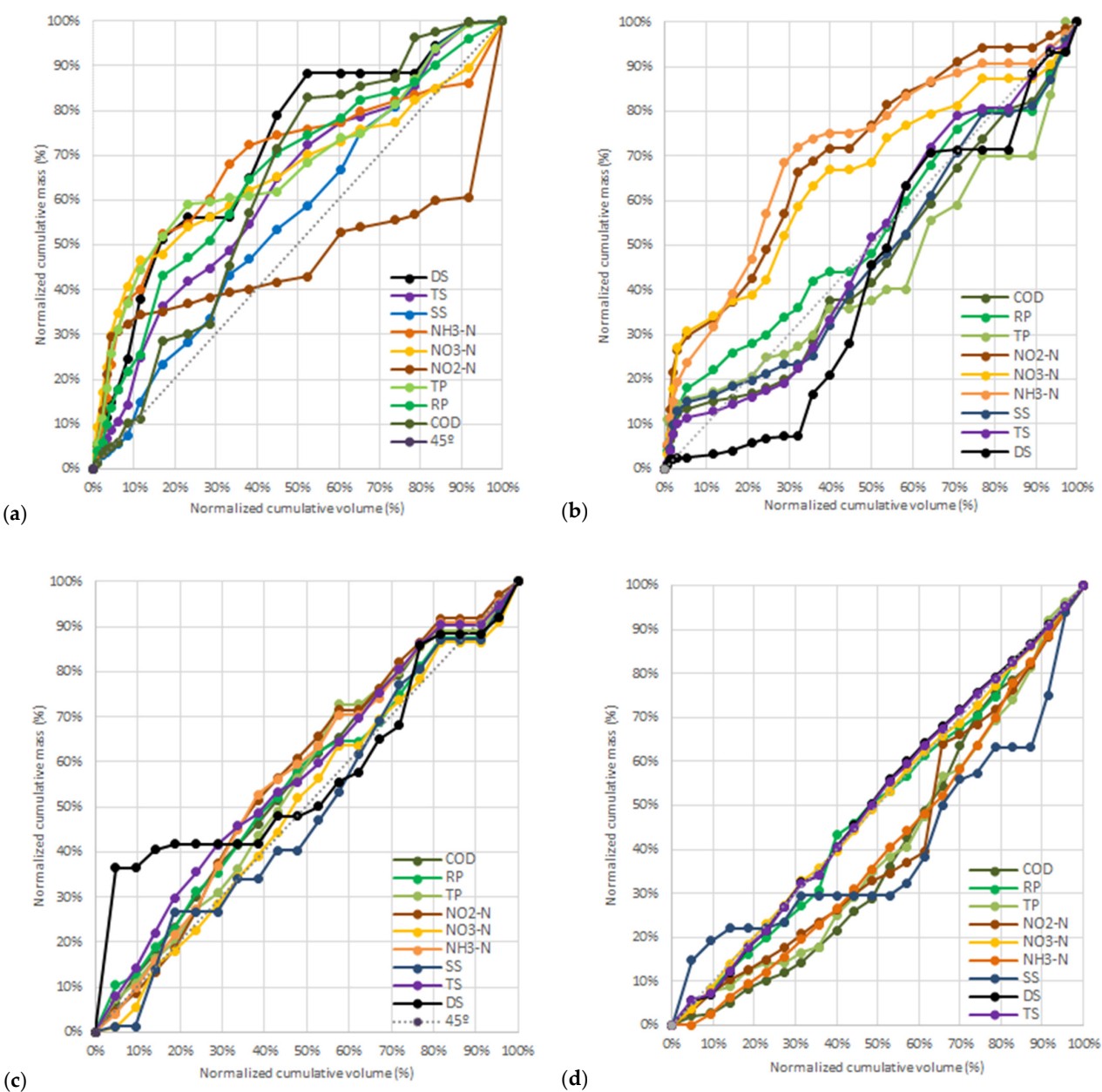

**Figure 5.** Cont.

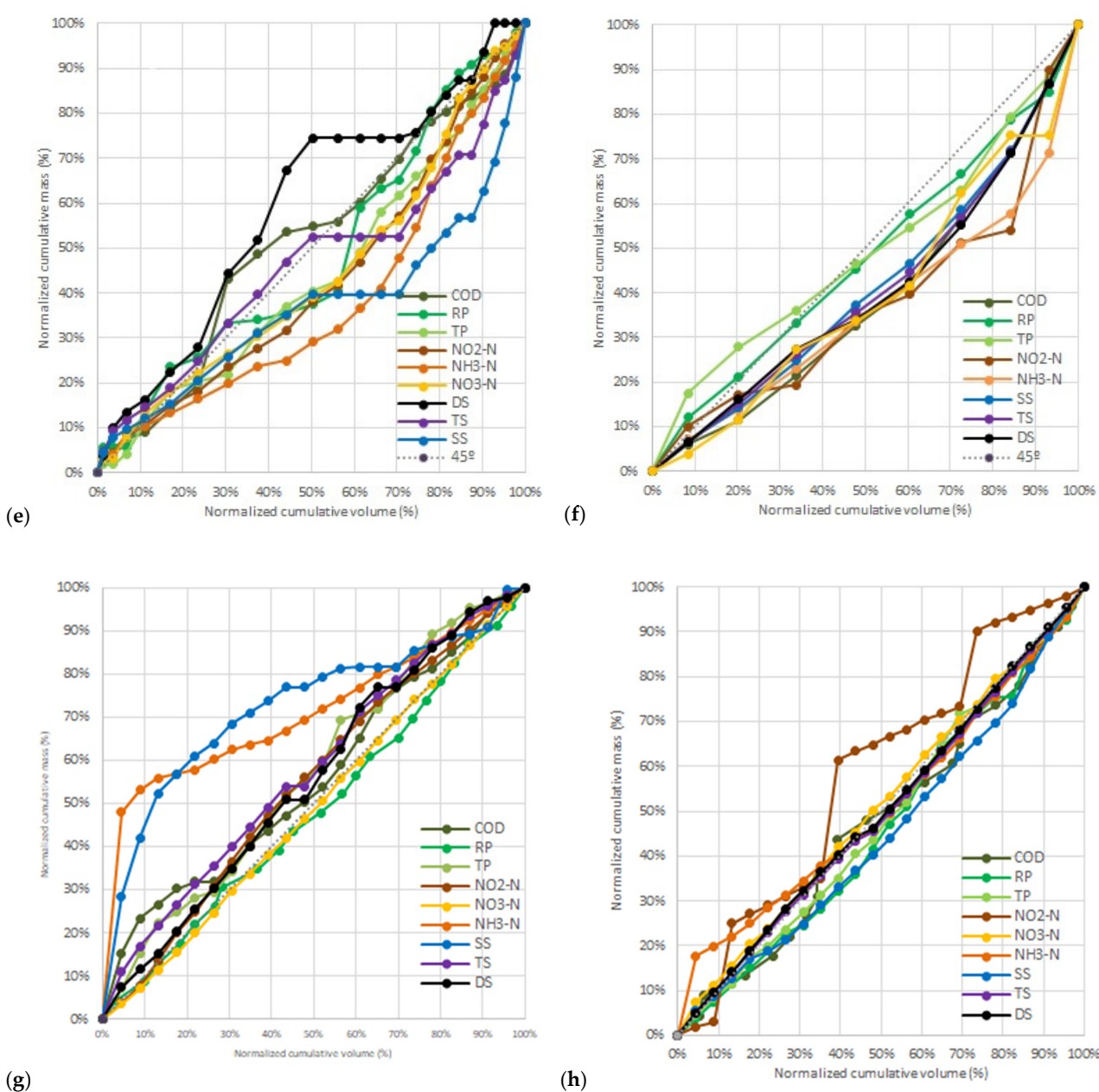

**Figure 5.** *Cont.*

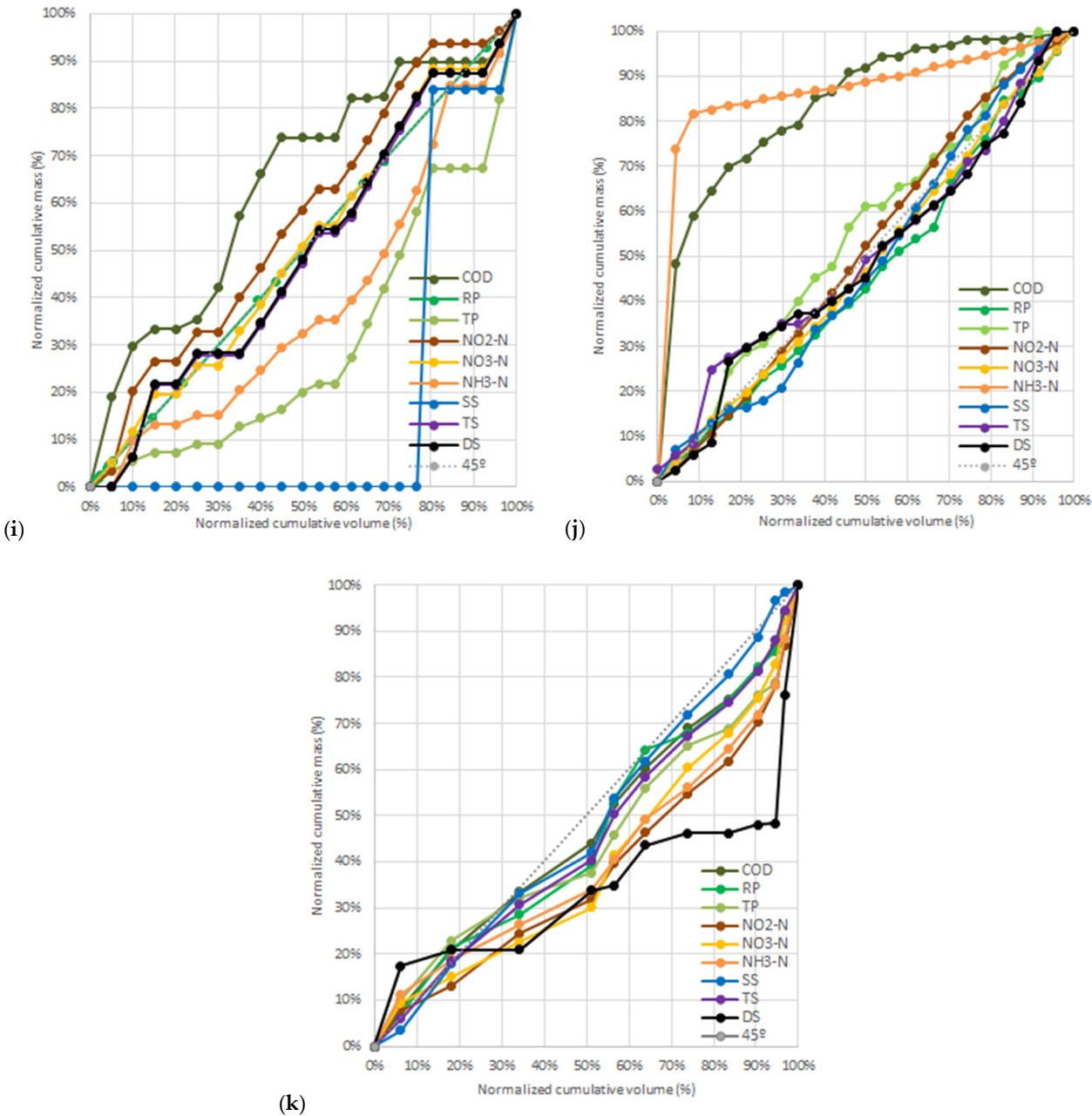

**Figure 5.** First flush effect during each event: (**a**)19 February 2018; (**b**) 4 March 2018; (**c**) 13 March 2018; (**d**) 1 August 2018; (**e**) 15 November 2018; (**f**) 18 February 2019; (**g**) 1 June 2019; (**h**) 19 June 2019; (**i**) 12 August 2019; (**j**) 29 August 2019; (**k**) 2 January 2020.

Bertrand-Krajewski et al. (1998) [49] determine the identification of the phenomenon with the 80/30 fraction. In this case, only the event that occurred on 29 August 2019 would present FF for two pollutants: $NH^{+3}$-N and COD, which limits the definition of the first flush. According to the definition of [46], first flush occurs when the M(V) curves are above the bisector, which would result in the identification of FF in all monitored events. When analyzed for parameter b, the occurrence of FF is confirmed, as b was less than 1 in 20 of the 45 curves (44%) for events monitored in the dry season, and 42 of the 54 curves (77%) for events monitored in the flood season.

Di Modugno et al., (2015) [63] reported that in the monitored events an average of the first 30% of the volume carries approximately 60% of SS. In the present research, what the results showed about FF was: for the first 30% of the initial volume, around 29.33% of SS

were carried during flood events (Table 4), and this is the lowest percentage. For NH$^{+3}$-N and RP, the greatest portions were 38.33% and 35%, respectively.

**Table 4.** Percentage of the mass of pollutants discharged in the first 30% of the volume of each analyzed event.

| Event | PR | PT | NO$_2$-N | NO$_3$-N | NH$_3$-N | SS | ST | SD | DQO |
|---|---|---|---|---|---|---|---|---|---|
| 19 February 2018 | 50 | 60 | 38 | 55 | 60 | 34 | 44 | 55 | 32 |
| 4 March 2018 | 34 | 26 | 60 | 52 | 70 | 28 | 20 | 7 | 20 |
| 13 March 2018 | 35 | 29 | 40 | 30 | 37 | 27 | 42 | 42 | 37 |
| 15 November 2018 | 33 | 22 | 24 | 26 | 20 | 37 | 39 | 44 | 43 |
| 18 February 2019 | 32 | 30 | 19 | 22 | 20 | 21 | 22 | 23 | 20 |
| 2 January 2020 | 26 | 30 | 21 | 21 | 23 | 29 | 28 | 21 | 30 |
| Average | 35.00 | 32.83 | 33.67 | 34.33 | 38.33 | 29.33 | 32.50 | 32.00 | 30.33 |

The values for b and the hydrological parameters, rainfall and flow rate, were correlated to assess whether a greater intensity of rainfall would influence the process of removal of soil particles and in the magnitude of the diffuse pollution, as stated by [64,65]. Analyzing Figures S1 and S2, found in the supplementary files, it can be seen that the greater the maximum or mean rainfall intensity is, the lower are the values of b for most pollutants, which was also identified by [66–69]. The only exception was DS, which can be associated with dilution issues.

It is perceived that the greater the number of antecedent dry days is, the lower is the value of b (that is, the greater is the occurrence of FF) for some pollutants, such as RP, NH$_3$-N, SS, and COD. This was also identified by [8] for the RF I watershed, where the event that had the lowest values of b also had the greater number of antecedent dry days, around 25 days. As confirmed by [64], the lowest values of b occurred more frequently after a long antecedent dry period, and according to [69] the number of antecedent dry days and the maximum rainfall intensity are the parameters that most affect the occurrence of FF.

Regarding the flow parameters, it is noted that the greater the peak flow and the flow volume are, the harder it is to characterize FF. This occurred in the event with the greatest flow rate, in 18 February 2019, which can be attributed to the great stormwater volumes in the event, causing a greater drag force, and also more dilution of the pollutants.

Despite the various, and sometimes ambiguous, definitions of first flush, researchers, policymakers, and watershed stakeholders widely recognize the potential of the knowledge about FF to promote more effective and economic implementation of practices to control river water quality, such as structures built with green infrastructure technology, since compensatory techniques for urban drainage have a limitation regarding the volumes they can retain [10,70–72].

### 3.4. Managerial Insights

The assessment of water quality in the VP River resulted in a real dataset that can support managers in the decision-making process. Some assumptions that can be drawn from the results of this research regarding stormwater management and water resources are:

1. The monitoring of water bodies during flood events is fundamental for the analysis of parameter compliance to quality standards and should be incorporated into water resource management plans;
2. Correlations between water quality parameters can be used to simplify monitoring when continuous monitoring is needed, but the subject may require more in-depth studies for each specific situation;
3. Investments in better sewage collection/treatment and urban water management systems are essential for mitigation of the environmental impacts of effluents, but they must be carried out paired with the urban expansion and always include measures to reduce the construction environmental impacts;

4.  Structures planned to retain the first volumes of runoff can be important for improving the quality of receiving water bodies, but must be hydraulically well designed and engineered;
5.  SUDS should be designed and implemented from the beginning of new urban settlements, but adapting existing stormwater management systems to become more sustainable is a viable option for controlling water pollution in the Federal District.

## 4. Conclusions

Based on the various parameters evaluated, this study concluded that the physical and chemical characteristics of the river water associated with rainfall and runoff data can characterize the diffuse pollution generated in the VP watershed, an area undergoing a rapid urbanization process in Brasilia, Brazil. In addition, the *EMC* values and the ratios between volume and pollutant mass transported found led to some conclusions about the behavior of pollutant concentrations in the VP river and the occurrence of the first flush phenomenon.

It was observed that the seasonal distribution pattern of different parameters was influenced by different anthropic factors. In the dry season, the concentration of organic matter and nutrients is higher due to the continuous discharge of untreated sewage into the river. In the rainy season the phenomenon of erosion stands out, being attributed to the large volumes of runoff, and the concentrations of nutrients and organic matter in the river are still high during rainy events, possibly induced by the overflow of septic tanks. These issues are associated with the rapid expansion of settlements linked to the lack of effective land-use management strategies.

The installation of sewage collection and treatment and urban drainage systems aims to reduce nutrient and organic matter pollution in the VP River, but in the monitored flood events, high sediment loads produced in the watershed were found, which are associated with the construction of these urban infrastructure networks that still takes place today, with constant excavation and earthworks resulting in high concentrations of SS. Therefore, the construction of the sewage and stormwater collection networks caused a high impact on the observed water quality, but on the other hand with the wastewater collection provided by the installation of the sewage network a decrease in COD concentrations was noticed over time in the dry period.

Compared to other studies, the *EMC* values found for the VP watershed were lower than those of two other Brazilian urban watersheds for ammonia and COD, but similar to those found in a small urban watershed in Brazil during flood events for COD and to those found in an area under construction also in the rainy season for SS.

Confirmation of the occurrence of FF in the VP watershed and the calculated volume-to-mass ratios for the pollutants help establish directions for sizing Low Impact Development devices used to achieve pollutant reduction goals, improve sampling methodologies for water quality assessment, and support the delineation of stormwater management strategies by public authorities.

The correlation values found in the study showed a significant increase/decrease of one parameter over the other. Rainfall events cause large floods that wash off pollutants through impervious urban areas. Large rainfall volumes generate high *EMC* values for ammonia, nitrate, SS, and COD; the latter two are also associated with high peak flows. Correlation analysis also showed that the higher the rainfall intensity and the number of ADDs, the higher the occurrence of FF for reactive phosphorus, ammonia, SS, and COD.

Finally, it was possible to identify management insights arising from the results of the study, mainly related to monitoring techniques and pollution control measures. In summary, the analysis of the *EMC* and FF and the correlation between them and the precipitation and runoff characteristics of monitored events provide valuable information for the management of water resources in the Federal District, in particular with regard to water pollution control in the VP river basin, and highlights the importance of continuous

monitoring of water quality in rainy periods for the purposes of stormwater management and diffuse pollution mitigation.

**Supplementary Materials:** The following are available online at https://www.mdpi.com/article/10.3390/w13182552/s1, Figure S1: Correlation between the hydrological and the FF characteristics during dry events, Figure S2: Correlation between the hydrological and the FF characteristics during rainy events.

**Author Contributions:** Conceptualization: M.E.L.C.; Methodology: M.E.L.C. and S.K.; Monitoring and Modelling: M.E.L.C. and D.J.C.; Formal Analysis: M.E.L.C. and S.K.; Writing—Original Draft Preparation: D.J.C. and M.E.L.C.; Writing—Review & Editing: S.K.; Supervision: S.K. All authors have read and agreed to the published version of the manuscript.

**Funding:** This research was funded by Brazilian Coordination for the Improvement of Higher Education (CAPES) and ANA (Sanitation and Water National Agency). The Brazilian National Council for Scientific and Technological Development (CNPq), the Federal District Research Support Foundation (FAP-DF), and the University of Brasília (UnB).

**Acknowledgments:** We acknowledge to FAPDF, FINEP, CNPq, FUNAPE, The authors are also grateful to ANA, Caesb, Novacap, and Adasa for data collection support.

**Conflicts of Interest:** The authors declare no conflict of interest.

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
