# Peer review of "Assessment of Pollutants from Diffuse Pollution through the Correlation between Rainfall and Runoff Characteristics Using EMC and First Flush Analysis"

_water, doi:10.3390/w13182552_

Round 1

Reviewer 1 Report

The authors presented a manuscript entitled "Assessment of Diffuse Pollution from a highly urbanized wa-tershed in Brazil". After the evaluation I have to share the comments bellow:

1) The article English needs to undergo significant improvement;
2) Abstract needs improvement to include quantitative results;
3) The keywords do not reflect the work! Relevant keywords are missing;
4) The Introduction section is extremely poor! This section is very limited and just roughly describe the situation of the study area, which is not the purpose of an introduction. You should rather present a (i) background of the research in this area (state of the art based on relevant recent research on the field), (ii) research rationale, (iii) research scope, (iv) research questions/hypothesis, (v) main research innovation, and (v) objectives;
5) Thes Section 2.2 (Field data acquisition system) is also methodology of the entire research? You do not have a section dedicated to methods! I was expecting a section 2.3 (Methods) with full description of the research approach. The "research" work performed does not justify a journal article. Therefore, there is no way to improve this article in order to make it worthy of publication;
6) The section 3 (Results and Discussion) is very poor and contains information that should be included in a section dedicated to methods. Figures are also of bad quality;
7) Conclusions are just speculation and are not related with the "research findings", supported by the manuscript content.

Author Response

Thank you for all contributions and the opportunity to resubmit this article for this important journal.

Point 1: The article English needs to undergo significant improvement;

Response 1: ok, we improved the English language with revision. We do not have time enough to submit to mdpi language revision, if you think it is necessary we can do for the next step.

Point 2: Abstract needs improvement to include quantitative results;

Response 2: ok, we added the quantitative results and some information more as details about introduction, methodology and conclusion.

Point 3: The keywords do not reflect the work! Relevant keywords are missing;

Response 3: Other keywords are added, more reflective about the paper.

Point 4:  4) The Introduction section is extremely poor! This section is very limited and just roughly describe the situation of the study area, which is not the purpose of an introduction. You should rather present a (i) background of the research in this area (state of the art based on relevant recent research on the field), (ii) research rationale, (iii) research scope, (iv) research questions/hypothesis, (v) main research innovation, and (v) objectives;

The research gap should be adequately explained.

In the introduction, please rearrange/rewrite so that each authors’/most of the authors' contributions should be linked. Would you please try to maintain the literature sequentially?

The introduction should be based on the exact research gap. The literature review should be based on the specific keywords-based review. Finally, make an author's contribution table to show the novelty and effectiveness of the study.

Response 4:  We added literature review to provide a background, state of art and some information the research as innovation and scope. We tried to add all the suggestions for points I, ii, iii, iv and v.

We explained the gaps about the research.

We put the main contribution about some keyword in the introduction.

And we tried to change the sequence to maintain the a better fluidity and understanding the introduction.

We added the section 3.4 Managerial insights with the effectiveness of the study.

Point 5: The Section 2.2 (Field data acquisition system) is also methodology of the entire research? You do not have a section dedicated to methods! one reviewer was expecting a section 2.3 (Methods) with full description of the research approach.

Response 5: We divided the methodology, to propose a better understood about the research.

We improve the methodology putting the methods about water quality concentration used.

Point 6: The section 3 (Results and Discussion) is very poor and contains information that should be included in a section dedicated to methods. Figures are also of bad quality;

Response 6: We improve the information and change the place to methodology as it was recommended. The figures about FF is better resolution, and the figures 7 and 8 were added as Supplementary Material, because they are very big.

Point 7: Please write proper managerial insights to show the industry managers' benefit from this research and compare this study with these research studies for wastewater treatment: A sustainable development framework for a cleaner multi-item multi-stage textile production system with a process improvement initiative; An interactive fuzzy programming approach for a sustainable supplier selection under textile supply chain management" theoretically and methodologically the applicability of the proposed research.

Response 7: We add a section 3.4 with material insights as suggested.

 Point 8: Would you please write the significant findings in conclusions? Do not mention all assumptions which have been indicated within the model.

Response 8: We rewrite the conclusion with phases more directs, to summarise the significant conclusions.

Reviewer 2 Report

  1. The abstract and conclusions are very short.  The way of conceptual writing is not perfect. The abstract should contain the details of the study and the findings in a very constructive way. The abbreviation should not be in the abstract. If needed, it can be started from the introduction onwards. The conclusions should be extended with significant findings and limitations. The applicability of the model should be explained.
  2. The research gap should be adequately explained.
  3. In the introduction, please rearrange/rewrite so that each authors’/most of the authors' contributions should be linked. Would you please try to maintain the literature sequentially?
  4. The introduction should be based on the exact research gap. The literature review should be based on the specific keywords-based review. Finally, make an author's contribution table to show the novelty and effectiveness of the study.
  5. Please write proper managerial insights to show the industry managers' benefit from this research and compare this study with these research studies for wastewater treatment: A sustainable development framework for a cleaner multi-item multi-stage textile production system with a process improvement initiative; An interactive fuzzy programming approach for a sustainable supplier selection under textile supply chain management" theoretically and methodologically the applicability of the proposed research.
  6. Would you please write the significant findings in conclusions? Do not mention all assumptions which have been indicated within the model.

Author Response

Thank you for all contributions and the opportunity to submit this paper for this important journal.

Point 1: The abstract and conclusions are very short.  The way of conceptual writing is not perfect. The abstract should contain the details of the study and the findings in a very constructive way. The abbreviation should not be in the abstract. If needed, it can be started from the introduction onwards. The conclusions should be extended with significant findings and limitations. The applicability of the model should be explained.

Response 1: we added the quantitative results and some information more as details about introduction, methodology and conclusion. We explained the abbreviations.

Point 2:; The research gap should be adequately explained.

Response 2: In introduction we put about the gap of the research.

Point 3:In the introduction, please rearrange/rewrite so that each authors’/most of the authors' contributions should be linked. Would you please try to maintain the literature sequentially?

Response 3: We changed the sequency and added more information about reviewr literature and state of art the topic in the paper.

Point 4:  The introduction should be based on the exact research gap. The literature review should be based on the specific keywords-based review. Finally, make an author's contribution table to show the novelty and effectiveness of the study.

Response 4: We based the introduction in the rewire to suppot knowlgde enough to the paper. We did not do the table with the author's contribution because we though the news paraghaphs supplied the lack of information.

Point 5: Please write proper managerial insights to show the industry managers' benefit from this research and compare this study with these research studies for wastewater treatment: A sustainable development framework for a cleaner multi-item multi-stage textile production system with a process improvement initiative; An interactive fuzzy programming approach for a sustainable supplier selection under textile supply chain management" theoretically and methodologically the applicability of the proposed research.

Response 5: We did in the section 3.4, based in the articles suggested.

Point 6: Would you please write the significant findings in conclusions? Do not mention all assumptions which have been indicated within the model.

 Response 6: We rewrite the conclusion based on the suggestions.

Reviewer 3 Report

Title:  Assessment of Diffuse Pollution from a highly urbanized watershed in Brazil

It’s a good topic. 

But the manuscript needs major revisions to improve the quality of the paper.

Abstract

Good abstract.

Introduction

Good introduction.

However, it can be expandable with more recent literature in this field by introducing the new references.

Methods and materials

Study area

More physical and fiscal geographical information is need to expand the study area. Also justification needed why did area has been selected for the assessment.

Methodology

Straightforward methodology.

Results

Well written section.

Figure 5, 6 and 7 has too many sub figures. No need of that much of figures. Revisions needed.

Conclusion

Should be in one paragraph.

What are the recommendation?

The authors should need to insert the limitations part.

Author Response

Thank you for all contributions and the opportunity to submit this paper for this important journal.

Point 1: Title:  Assessment of Diffuse Pollution from a highly urbanized watershed in Brazil

It’s a good topic.

But the manuscript needs major revisions to improve the quality of the paper.

Response 1: Thanks. We tried to explore this topic at the study area. We did major revisions to provide more quality for the paper.

Point 2: Introduction

Good introduction.

However, it can be expandable with more recent literature in this field by introducing the new references.

Response 2: We expanded the review literature and stat of art about the them.  We also the change the organization to give more fluidity to the text.

Point 3: Study area

More physical and fiscal geographical information is need to expand the study area. Also justification needed why did area has been selected for the assessment.

Response 3: We added the information requested. We said the motive the choice of the study area, mainly because the use and occupation and the fast urbanization with infrastructure construction.

Point 4:  Methodology

Straightforward methodology.

Response 4: We improve the explanation about the methodoghy adopted. We provided more information about the methods and divided the tops to get more understandable.

Point 5: Results

Well written section.

Figure 5, 6 and 7 has too many sub figures. No need of that much of figures. Revisions needed.

Response 5: We changed all the figures to get better resolution. The figures 6 and 7 were moved to Supplementary Material.

Point 6: Conclusion

Should be in one paragraph.

Response 6: Others reviews asked more information about conclusion. So we tried to get more direct with the main benefits with this research.

Round 2

Reviewer 1 Report

The authors did not give necessary attention to my recommendations. The manuscript is still of very low quality and does not bring any contribution.

The last paragraph of the section 4 (conclusions) express the low quality of this article and the absence of contribution to the consolidation of knowledge, even in the perspective of the study area:

"It has been concluded that to understand non-point source phenomena continuous monitoring of flow and water quality variables is required. This will help with understanding the physical processes accurately and enhance the capability of mathematical models. Besides that, assist to develop a strategy to manage the environmental hazards due to pollution and to improve environmental protection of the VP River."

Author Response

Dear eviewer,

We are grateful for the valuable contributions made to the manuscript and are glad to have been able to meet several of your recommendations. The text has been revised again to correct any grammatical errors and to check the spelling.

The authors did not give necessary attention to my recommendations. The manuscript is still of very low quality and does not bring any contribution.

The last paragraph of the section 4 (conclusions) express the low quality of this article and the absence of contribution to the consolidation of knowledge, even in the perspective of the study area:

"It has been concluded that to understand non-point source phenomena continuous monitoring of flow and water quality variables is required. This will help with understanding the physical processes accurately and enhance the capability of mathematical models. Besides that, assist to develop a strategy to manage the environmental hazards due to pollution and to improve environmental protection of the VP River."

We apologize for not meeting your expectations regarding the corrections.

We have tried once again to improve the questioned points, including the conclusion. We have reorganized section 4 (Conclusions) and rewritten part of it, with emphasis on the correlations between water quality parameters and characteristics of the events monitored and their potential use in assisting stormwater management. We would like to highlight the fact that the study was developed in an area with no availability of monitored water quality data during flood events, and we believe that the region, which suffers from severe water pollution problems, mainly linked to runoff during rainfall events, can benefit from the results.

We thank you again for your attention in reviewing this manuscript.

Best regards,

The authors.

Reviewer 2 Report

The paper can be accepted for publication.

Author Response

Dear Editors and Reviewer,

We are grateful for the valuable contributions made to the manuscript and are glad to have been able to meet several of your recommendations. The text has been revised again to correct any grammatical errors and to check the spelling. The reference list was also checked for missing citations and standardized according to MDPI guidelines.

Aiming to improve the retrieval of the results found to support the conclusions drawn and better express the contributions brought by the study, we have reorganized section 4 (Conclusions) and rewritten part of it, with emphasis on the correlations between water quality parameters and characteristics of the events monitored and their potential use in assisting stormwater management. We would like to highlight the fact that the study was developed in an area with no availability of monitored water quality data during flood events, and we believe that the region, which suffers from severe water pollution problems, mainly linked to runoff during rainfall events, can benefit from the results.

We thank you again for your attention in reviewing this manuscript.

Best regards,

The authors.

Reviewer 3 Report

Authors addressed all of my comments. 

Author Response

(The authors gave the same response as above.)
